# Real-Time Detection of Drones Using Channel and Layer Pruning, Based on the YOLOv3-SPP3 Deep Learning Algorithm

**DOI:** 10.3390/mi13122199

**Published:** 2022-12-11

**Authors:** Xuetao Zhang, Kuangang Fan, Haonan Hou, Chuankai Liu

**Affiliations:** 1School of Mechanical and Electrical Engineering, Jiangxi University of Science and Technology, Ganzhou 341000, China; 2Key Laboratory of Magnetic Levitation Technology in Jiangxi Province, Ganzhou 341000, China; 3School of Electrical Engineering and Automation, Jiangxi University of Science and Technology, Ganzhou 341000, China; 4Ganjiang Innovation Academy, Chinese Academy of Sciences, Ganzhou 341000, China; 5Beijing Aerospace Control Center, Beijing 100000, China; 6National Key Laboratory of Science and Technology on Space Flight Dynamics, Beijing 100000, China

**Keywords:** anti-drone, feature extraction, real-time detection, YOLOv3-SPP3, pruning algorithm

## Abstract

Achieving a real-time and accurate detection of drones in natural environments is essential for the interception of drones intruding into high-security areas. However, a rapid and accurate detection of drones is difficult because of their small size and fast speed. In this paper a drone detection method as proposed by pruning the convolutional channel and residual structures of YOLOv3-SPP3. First, the k-means algorithm was used to cluster label the boxes. Second, the channel and shortcut layer pruning algorithm was used to prune the model. Third, the model was fine tuned to achieve a real-time detection of drones. The experimental results obtained by using the Ubuntu server under the Python 3.6 environment show that the YOLOv3-SPP3 algorithm is better than YOLOV3, Tiny-YOLOv3, CenterNet, SSD300, and faster R-CNN. There is significant compression in the size, the maximum compression factor is 20.1 times, the maximum detection speed is increased by 10.2 times, the maximum map value is increased by 15.2%, and the maximum precision is increased by 16.54%. The proposed algorithm achieves the mAP score of 95.15% and the detection speed of 112 f/s, which can meet the requirements of the real-time detection of UAVs.

## 1. Introduction

With the rapid development of drone technology in recent years, low-cost small drones have gradually become popular and have been widely used in various fields. Despite bringing convenience to people’s lives, these drones also introduce a series of problems, such as breaking into no-fly zones, crashing into high-voltage power lines, and disrupting navigation. While the rapid development of the civil drone industry has brought convenience to people’s lives, it has also posed a great risk to public safety and has had a serious negative social impact. Although there are a series of laws and regulations for civil drones, drones are small devices that fly in the air, and the legal awareness of drone operators is low, so it is extremely difficult to regulate them, leading to a variety of illegal and irregular incidents, “black flying”, and “indiscriminate flying “, resulting in incidents that affect the social security and stability [1,2]. For example, in 2019, unidentified drones intruded into a military zone in Wuhan, China, spying on the secrets of the military zone. in 2021, drones were found near train tracks in areas, such as Xi’an and Changsha, China. The “indiscriminate” flights affected the safe operation of trains. So, there is great value in researching the detection and identification of drones.

When entering laboratories, military, private, and other areas, these drones introduce great security risks to the country and its people. Therefore, anti-UAV systems [3,4] should be adopted to address these problems. According to Wu et al., in order to improve the accuracy and speed of the image target detection under complex weather conditions, an image target detection method, based on the YOLOv3 algorithm and image enhancement processing is proposed [5]. According to Li and Liang, in order to speed up the detection speed of the algorithm, the YOLOv3 algorithm model is simplified by quantization pruning, and the training of the YOLOv3-tiny algorithm model is completed [6].

Some studies use acoustic signals to detect drones [7,8]. Some systems use the fast Fourier transform to convert the sampled real-time data and then perform drone detections using the transformed data along with two methods, namely, the plotted image machine learning and the k-nearest neighbor (KNN) [9]. The position of the drone is determined via the short-time frequency domain beams. Based on the detected positions, an α-β filter was used to reconstruct the individual trajectories [10]. Albeit simple, detecting drones by acoustic signals covers a limited distance and is prone to interference. Radars are also commonly applied in object detection. An iterative adaptive method was used to perform doppler processing in ground-based surveillance radars for drone detection [11]. While the radar returns good detection results, this approach is not suitable for densely populated areas and has a high price.

Additionally, in the early days of object detection, motion-based algorithms, such as optical flow [12], inter-frame difference, and background subtraction [13], were applied, and then a support vector machine [14] was used to classify the results. The histogram of the oriented gradient method [15,16] was also utilized for classification and detection. The traditional target detection algorithm, based on the manual feature extraction, mainly has some shortcomings, such as a poor recognition effect, a low accuracy, a large amount of computation, a slow operation speed, and may produce multiple correct recognition results.

Compared with traditional methods, a convolutional neural network (CNN)—based target detection [17,18,19] and recognition methods demonstrate a better performance. In recent years, with the rise of the CNN and improvements in computer computing power, object detection algorithms, based on deep learning, have rapidly developed. CNNs can effectively extract image features. Since the R-CNN [20] was introduced in 2014, many state-of-the-art deep target detection algorithms have been proposed, which can be categorized into two- and single-stage detection algorithms.

Two-stage detection algorithms, including SPP-net [21], fast R-CNN [22], faster R-CNN [23], and R-FCN [24], have been designed in recent years. These algorithms comprise a backbone network, a region proposal module, and a detection header. They generate proposal regions using the region proposal module to achieve a good detection accuracy. SPP needs to train the CNN to extract features, and then train the SVM to classify these features, which requires a huge amount of storage space, and the multi-stage training process is very complicated. In addition, SPP-net only fine-tunes the fully connected layer and ignores the parameters of the other layers of the network. The fast RCNN still selects the selective search algorithm to find the area of interest, which is usually a slow process. Although the faster RCNN has a higher accuracy, a faster speed and is very close to the real-time performance, it still has some computational redundancy in the subsequent detection stage. In addition, if the IOU threshold is set low, the noise detection will be caused; if the IOU threshold is set high, overfitting will be caused.

However, this method requires much calculation and running time, thereby resulting in a relatively slow detection. Given their end-to-end neural networks, the single-stage detection algorithms run faster than the two-stage detection algorithms. These algorithms, such as the YOLO series (YOLO [25], YOLOv2 (YOLO9000) [26], and YOLOv3 [27]), SSD [28], and RetinaNet [29], utilize predefined anchor points instead of a regional proposal network, to cover the image space position. YOLOv3 achieves a good balance between the detection accuracy and speed, thereby making this deep object detector popular in practical applications. However, when applied to embedded devices, YOLOv3 requires a large computational overhead and power consumption. Compared with the two-stage target detection algorithm, although the detection speed of the YOLOv1 algorithm has been greatly improved, the accuracy is relatively low. The YOLOv2 algorithm has only one detection branch, and the network lacks the capture of the multi-scale context information, so the detection effect for the targets of different sizes is still poor, especially for the small target detection. The detection speed of the SSD algorithm is slow and cannot meet the requirements of real-time detection.

Model compression methods include model pruning [30,31] parameter quantification [32] and knowledge distillation [33]. Ye, J. et al. [34] proposed a channel pruning technique to accelerate the computations of deep CNNs. This technique directly uses the scaling factor in batch normalization (BN) as the channel scaling factor. The channel with a large shift factor has a great influence on the convolutional layers. Yijie Chen et al. [35] proposed high-ratio channel pruning (HRCP). This method is based on the definition of the channel saliency with the scale factor and shift factor of the BN layer. Wu et al. [36] proposed slimming binarized neural networks (SBNNs), which reduce the complexity of binarized networks with an acceptable accuracy loss. Wang et al. [37] proposed a scheme of a network channel pruning, based on sparse learning and the genetic algorithm. SlimYOLOv3 [38] removes unimportant convolution channels by pruning the convolution channels of YOLOv3-SPP3, thereby reducing the number of trainable parameters and floating-point operations. Although YOLOv4, YOLOv5, and YOLOv6 are later versions of YOLOv3, they do not abandon the original network of YOLOv3 and still adopt the overall network combining the backbone network with the multiscale feature extraction network. YOLOv3 still has high research value. In the Internet of Things (IoT) scenario, the integration with cloud-based solutions is crucial to address the shortcomings caused by resource-constrained issues. A FPFTS fog task scheduler using particle swarm optimization and fuzzy theory is presented in the literature, which exploits the observations related to the application loop latency and the network utilization [39]. The Internet of Drones (IoD), with its wide range of promising applications, such as aerial photography, civil, and military purposes, is comprehensively described in the literature, along with its applications, deployment, and integration [40]. Potential technologies and applications for collaborative drones and the Internet of Things, that were recently proposed to enhance the intelligence of smart cities, are presented in the literature and provide a comprehensive overview, highlighting the recent and ongoing research on collaborative drones and the IoD, for improving smart city applications in real time [41].

In order to solve the problems of low accuracy and low efficiency in the detection and recognition of small targets of unmanned aerial vehicles (UAVs), we proposed a fast identification and detection method for UAVs, based on the pruning of the YOLOv3-SPP3 convolution channel and residual structure (YOLOv3-SPP3 algorithm). Firstly, the K-means algorithm was used to cluster the label boxes, and then the channel pruning and layer pruning algorithms were used to prune the model. Finally, the model was fine-tuned. YOLOv3-SPP3 uses Darknet-53 as its backbone network, which has 23 fast network layers. The model is further compressed by trimming the unimportant convolution channels and the fast network layer. The experimental results show that, compared with five reference algorithms, namely YOLOv3, Tiny-YOLOv3, CenterNet, SSD300 and faster R-CNN, it has a good performance in precision, recall, model size, and detection speed, and has excellent characteristics, such as real-time, rapidity, and accuracy.

## 2. Related Works

In order to successfully detect and recognize small UAVs, we first carried out the preparation of the UAV dataset, and after the dataset was annotated, the sample features of the UAV dataset were re-clustered using the K-means clustering algorithm; and the YOLOv3-SPP3 algorithm was proposed.

### 2.1. UAV Image Acquisition

DJI spark and phantom were used as subjects. The images were collected from Jiangxi University of Science and Technology. A Honor 10 mobile phone was used to collect the drone videos from different angles and distances. The frame rate was set to 30 fps, and the videos were saved in MP4 format. The collected drone videos were decomposed into image sequences using the Opencv image processing package. Given the great similarity in the image sequences between neighbors, an image was taken every five frames. Samples of drone images are shown in Figure 1.

Table 1 summarizes the information on the UAV datasets. A total of 8487 drone images with a resolution of 1920 pixels (horizontal) × 1080 pixels (vertical) were collected, among which, 3441 images include DJI spark, 3315 images include DJI phantom, and 1731 images include both DJI spark and phantom. LabelImg was used to manually label the drones in these 8487 images, and each drone was located at the center of the bounding box when labeled. A corresponding xml file containing the coordinates of the top left and bottom right corners of the drone was generated. Around 80% (6790 images) of the prepared datasets were used as training data, and the remaining 20% (1697 images) were used as test data.

### 2.2. K-Means Clustering of UAV Datasets 

In the original YOLOv3 algorithm, the K-means algorithm was used to cluster the labeled boxes in the VOC and COCO datasets, and nine anchor boxes with sizes of (10, 13), (16, 30), (33, 23), (30, 61), (62, 45), (59, 119), (116, 90), (156, 198), and (373, 326), were used. The VOC and COCO datasets included multiple types of targets. However, the UAV dataset used in this study had a small size. Therefore, re-clustering should be performed, according to the sample characteristics of the UAV datasets, to obtain new anchor boxes. The following distance measurement formula was used in the K-means clustering:

*d*(*box,centroid*) = 1 − (*IoU*(*box,centroid*))
(1)


With K = 9, the clustered results of the K-means algorithm on the UAV dataset labeled boxes are shown in Figure 2, and the sizes of the obtained nine new anchor boxes were (8, 18), (10, 30), (17, 14), (31, 32), (16, 40), (15, 26), (23, 47), (31, 70), and (102, 102).

### 2.3. Model Presentation—YOLOv3-SPP3

YOLO is a one-stage object detection algorithm that uses a simple CNN to process images and directly calculates the classification results and position coordinates of the objects. The backbone network used by YOLOv2 (YOLO9000) is DarkNet-19, which is similar to the VGG network. YOLOv3 adopts a new residual network, whose structure contributes to a gradual decline in the model gradient, thereby further enhancing the network depth and improving the model accuracy. In addition to using residual blocks, YOLOv3 refers to other methods, such as SSD, to introduce multi-scale feature maps and improve the target detection accuracy. Three feature maps with different scales were used. Each scale of these feature maps had three corresponding groups of anchor boxes. The test results reveal that this method effectively improves the recall rate for the small target objects and solves the defects in the first two YOLO versions.

Figure 3 shows the spatial pyramid pooling (SPP) network structure diagram, which comprises four parallel max-pool layers with kernel sizes of 1 × 1, 5 × 5, 9 × 9, and 13 × 13. The SPP module extracts features with different receptive fields, connects them to the channel dimension of the feature map, and blends them together. A SPP module was added between the 5th and 6th convolutional layers in front of each detection header in YOLOv3, and this model was renamed as YOLOv3-SPP3.

## 3. Method and Performance Evaluation

In this paper, four metrics, namely accuracy, recall, mAP, and detection speed, are used to validate the performance of the proposed target detection model.

In recent years, CNNs have been widely used in various studies and applications. With the continuous advancement in computer capabilities, the number of layers in the network increased from eight in AlexNet, to tens or even hundreds of layers. Convolutional neural networks with more layers have a greater expressive power yet require more memory space and computing resources. 

Despite accurately detecting drones, YOLOv3-SPP3 uses DarkNet-53 as its backbone network, which has a large number of network parameters. To improve the detection speed of the drones, the complexity of the model is reduced, and its future deployment on devices with weak computing capabilities, such as embedded devices, is promoted. Pruning the model is an indispensable process.

YOLOv3-SPP3 was pruned in this study. The adopted pruning strategy was divided into three stages, namely, sparse training, channel, and layer pruning, and model fine tuning. As is shown in the Figure 4.

The main parameters used in pruning YOLOv3-SPP3 are shown in Table 2.

### 3.1. Sparse Training

A sparse training of the deep CNN facilitates the pruning of less important convolutional channels and layers. To facilitate the channel and layer pruning, a scaling factor was assigned to each channel, where the absolute value of the scaling factors indicates the importance of the channels. The BN layer was used to accelerate the convergence of each convolutional layer in YOLOv3-SPP3. This layer can be formulated as Equation (2):
(2)$$y=\;\gamma \;\times \;\frac{x\;-\;\bar{x}}{\sqrt{\sigma^{2}+\varepsilon }}\;+\;\beta$$
where $\bar{x}$ and *σ*^2^ represent the mean and variance of the input features, whereas *γ* and *β* represent the trainable scaling factor and bias, respectively. Given that each convolutional layer in YOLOv3-SPP3 contains a BN layer, the trainable scaling factor *γ* was directly used in the BN layer as the channel importance indicator. To effectively distinguish the important channels from the unimportant ones, channel sparse training was performed by imposing L1 regularization on the scaling factor *γ*. The loss function in the sparse training was formulated as Equation (3):
(3)$$Loss=\;loss_{yolov3}+\alpha \sum_{\gamma \in \Gamma }f\left(\gamma \right)$$
where the first term is the loss of the network, and the second term is the *L*_1_ regular term of the *γ* parameters, where *α* is the balance parameter of two items. A larger *α* corresponds to a faster sparse training speed. The balance parameter *α* was set to 0.001. The scaling factor distribution of the BN layer before the sparse training is shown in Figure 5. Prior to the sparse training, the *γ* value of the BN layer follows a normal distribution.

The YOLOv3-SPP3 network is sparsely trained using the stochastic gradient descent method. The scaling factor distribution of the BN layer after the sparse training is shown in Figure 6. Following the training, training, most *γ* coefficients of the BN layer were approximately 0.1, with some reduced to 0. These smaller values denote the unimportant channels and can thus be pruned.

### 3.2. Channel and Layer Pruning

#### 3.2.1. Channel Pruning

Following the sparse training of YOLOv3-SPP3, a model with a *γ* scaling parameter close to zero in the BN layer can be obtained. An appropriate global threshold $\hat{\gamma }$ was set to determine whether the feature channel was pruned. A schematic diagram of the channel pruning algorithm is shown in Figure 7.

YOLOv3 contains 75 convolutional layers, of which the last three output convolutional layers do not have BN layers. Given that two of these convolutional layers are connected to the upsampling layer, the channels of these convolutional layers were not pruned. The remaining 70 convolutional layers used the global threshold $\hat{\gamma }$ to determine whether the channel was pruned. The pruning threshold affects the detection accuracy, model size, and inference time. 

To determine the optimal pruning rate of the convolution channel, when the loss of the mAP value is small, a relatively small model size and inference time were selected. The YOLOv3-SPP3 drone target detection model was tested under different channel pruning ratios, and the channel pruning ratio was defined as Equation (4):


(4)
$$\eta =\frac{\mathrm{Number}\;\mathrm{of}\;\mathrm{pruned}\;\mathrm{channels}}{\mathrm{Total}\;\mathrm{number}\;\mathrm{of}\;\mathrm{convolution}\;\mathrm{channels}}$$


In the experiment, the channel pruning ratio was set to 0, 0.1, 0.2, 0.3, 0.4, 0.5, 0.6, 0.7, 0.8, 0.9, 0.925, 0.95, and 0.98. The changes in the average accuracy, size, and inference time of the model, along with the increasing channel pruning are shown in Table 3. 

Figure 8 shows that when the channel pruning rate is below 0.8, the mAP value remains above 95%. When the channel pruning rate is 0.9, this value slightly decreases. When the channel pruning rate exceeds 0.9, this value rapidly decreases. The size of the model decreases along with an increasing channel pruning rate. Specifically, when the channel pruning rate is below 0.5, the model size rapidly decreases. When the channel pruning rate exceeds 0.5, the model size gradually decreases. Increasing the channel pruning rate also gradually reduces the inference time of detecting the UAV datasets. When the channel pruning rate exceeds 0.7, the inference time remains unchanged. 

The experiment results show that when the convolution channel pruning ratio is 0.9, the average accuracy of the pruned model is 93.24%, which is slightly lower than that of the original model. At the same time, the model size is 10.24 M, which is 95.7% smaller than the uncompressed model. Given that the inference time of the model was reduced from 20.3 ms to 12.2 ms, the pruning ratio of the convolution channel was set to 0.9.

#### 3.2.2. Layer Pruning

The YOLOv3-SPP3 drone detection model was designed, based on the YOLOv3 framework. As this network contains 23 residual structures (shortcut), in addition to the convolution channel, the convolutional layer in these residual structures can also be pruned. Two convolutional layers were present in the residual structure, and a convolutional layer was placed before the structure. Pruning a residual structure is equivalent to pruning three convolutional layers. The residual network includes BN layers, and the scaling factor in the BN layer was used to evaluate the importance of the residual network. All *γ* factors in the BN layer of the residual network were averaged, and the *γ* mean values of the 23 residual networks were sorted. The residual structure with a smaller scaling factor was pruned. 

The residual structure was pruned, based on channel pruning. In the experiment, the pruning numbers of the residual structure were set to 0, 2, 4, 6, 8, 10, 12, 14, 16, 18, and 20. The changes in the average accuracy, model size, and inference time of the model, along with increasing the residual structure are shown in Table 4.

As shown in Figure 9, when the number of pruned residual structures is below 12, the mAP score remains above 93%. When the number of pruned residual structures is 14, the mAP score slightly decreases. When the number of pruned residual structures exceeds 14, the mAP score rapidly decreases. When the number of residual structures is less than 12, increasing the number of pruned residual structures would slightly reduce the model size. When the number of residual structures exceeds 12, increasing the number of pruned residual structures would rapidly reduce the model size. The number of pruned residual structures is negatively correlated with the inference time. Increasing the number of pruned residual structures also reduces the inference time. 

The above experiments show that when the number of pruned residual structures is 14, the model obtains an average accuracy of 83.64%, which is slightly lower than that of the original model. At the same time, the model size is 9.64 M, which is 5.9% lower than before the residual structure pruning. The inference time of the model is also reduced from 12.2 ms to 8.7 ms after pruning the residual structure. Therefore, the number of pruned residual structures was set to 14. 

Following the channel and layer pruning, the number of convolution channels of YOLOv3-SPP3 is shown in Figure 10, where pink denotes the number of channels, and blue denotes the number of remaining channels. The number of channels in most convolutional layers was greatly reduced.

### 3.3. Model Fine-Tuning

The performance of the model was reduced after pruning. To compensate for the pruned information of the network and improve the target detection accuracy, the model was fine tuned to restore its previous performance as much as possible. Following the fine tuning, the model accuracy reached 95.15%, which is close to the original accuracy before pruning. The model size was 11.77 M, whereas the inference time was 8.9 ms.

Test results in Table 5 show that after pruning the model, the mAP score, size, and forward inference time were reduced by 12.28%, 229.47 M, and 11.6 ms, respectively. The model was fine tuned after pruning. Although the model size and inference time slightly increased after the fine tuning, its accuracy was close to that of the original detection model. The results indicate that pruning the channel and layer can simplify the model, improve the drone detection speed, and ensure the detection accuracy.

## 4. Experiment and Results

Learning efficient deep object detectors were developed in this study by pruning the less important feature channels and layers. The pruned YOLOv3-SPP3 model had fewer parameters and a lower computation overhead, compared with the original model. An Ubuntu server with 64 GB RAM, Intel(R) Xeon(R) CPU E5-2640 v4 @ 2.40 GHz, and 2 NVIDIA GTX1080 ti GPU cards was used for the evaluation, in addition to the Python 3.6 development environment and some necessary dependent libraries.

### 4.1. Evaluation Indicators

Four indicators, namely, accuracy, recall rate, mAP, and detection speed, were used to verify the performance of the proposed target detection model. Intersections over (IOU) values of 0.5, <0.5, and 0 represent true positive (TP), false positive (FP), and false negative (FN) cases, respectively.

(1) Precision 

The accuracy refers to the probability of the correct detection among all detected targets.


(5)
$$\mathrm{Precision}=\frac{TP}{TP+FP}$$


(2) Recall

The recall rate refers to the probability of the correct recognition among all positive samples. 


(6)
$$\mathrm{Recall}=\frac{TP}{TP+FN}$$


(3) mAP

The mAP is the average precision from the dimension of the category.


(7)
$$\mathrm{mAP}=\frac{AP}{\mathrm{num}\_\mathrm{classes}}$$


### 4.2. Comparison of the Different Object Detection Algorithms

To verify its performance in drone detection, the pruned YOLOv3-SPP3 model was compared with five object detection algorithms, namely, YOLOv3, Tiny-YOLOv3, CenterNet, SSD300, and faster R-CNN. 

On the basis of the target detection model of the above six algorithms, the UAV training set was used for the training, whereas the test set was used for evaluating each detection algorithm. Table 6 presents the test results.

A rapid and accurate detection of drones has an important role in anti-UAV systems. YOLOv3, Tiny-YOLOv3, CenterNet, SSD300, faster R-CNN, and the proposed method, were compared to determine their drone detection performance. The mAP scores of these algorithms were 95.27%, 79.95%, 96.21%, 85.42%, 98.96%, and 95.15%, the corresponding model sizes were 235.52, 33.79, 237.01, 91.45, 568, and 11.77 MB, and the detection speeds were 56, 145, 38, 49, 11, and 112 f/s, respectively.

The proposed algorithm outperforms all of the other five algorithms and achieves an excellent balance between speed and accuracy. In terms of the detection accuracy, although the mAP scores of YOLOv3, CenterNet, and faster R-CNN were higher than that of the proposed algorithm, the detection speed of the proposed algorithm can satisfy the real-time detection requirements. In terms of the detection speed, Tiny-YOLOv3 achieved a faster detection speed than the proposed method. However, the mAP score of the proposed algorithm was 15.2% higher than that of Tiny-YOLOv3. The proposed algorithm also had the smallest model size among all algorithms. The drone detection results obtained by this algorithm are shown in Figure 11. 

The first two rows of Figure 11 show images including DJI spark. The first row shows the drones with a long flying distance, whereas the second row shows the drones with a short flying distance. Meanwhile, the third and fourth rows present images that include the DJI phantom with near and far flying distances, respectively. The last row presents the results including both the DJI spark and DJI phantom. Figure 11 shows that all drones can be fully detected in various natural environments.

## 5. Conclusions

To achieve an accurate and real-time detection of drones in natural environments, this research proposed a drone detection method, based on the YOLOv3 and pruning algorithms. An accurate and rapid detection of drones was realized using the YOLOv3-SPP3 algorithm. The proposed method reduces the number of learnable parameters by pruning the YOLOv3-SPP3 channel and layers. Under the condition that the detection accuracy of the model is kept almost unchanged, the model size and inference time of the proposed drone detection model was reduced by 95.16% and 51.37%. The experimental results obtained with the Ubuntu server in the Python 3.6 environment show that the proposed method reduces the model size to 1/20, compared with YOLOv3 and doubles the detection speed; improves the map value by 15.2%, compared with Tiny-YOLOv3 and reduces the model size to 1/3, compared with Tiny-YOLOv3; improves the detection speed by three times, compared with CenterNet. Compared with Center Net, the detection speed is increased by three times, and the model size is reduced to 1/20 of the original size; compared with SSD300, the model size is reduced to 1/8 of the original size, and the detection speed is increased by 2.3 times; compared with faster R-CNN, the model size is reduced to 1/28 of the original size, and the detection speed is increased by 10 times. The proposed method can effectively simplify the detection model and realize an accurate and real-time detection of drones in natural environments.

## Figures and Tables

**Figure 1 micromachines-13-02199-f001:**
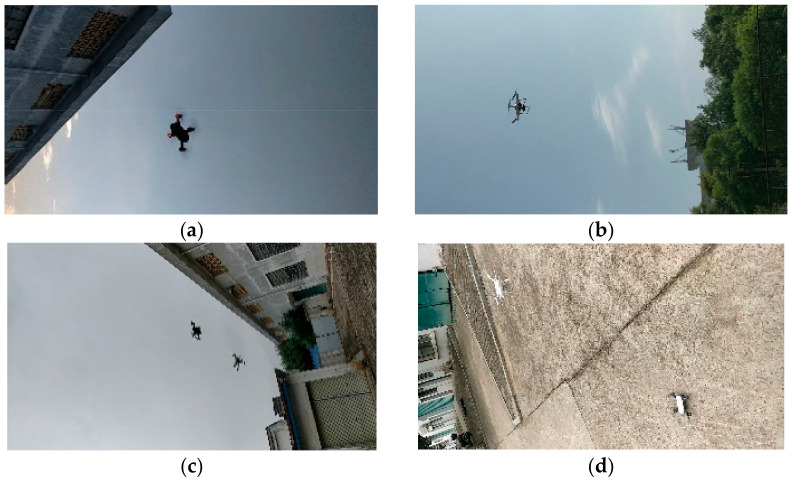
Examples from the UAV datasets: image (**a**) contains DJI spark; image (**b**) contains DJI phantom; images (**c**,**d**) contain both DJI spark and phantom.

**Figure 2 micromachines-13-02199-f002:**
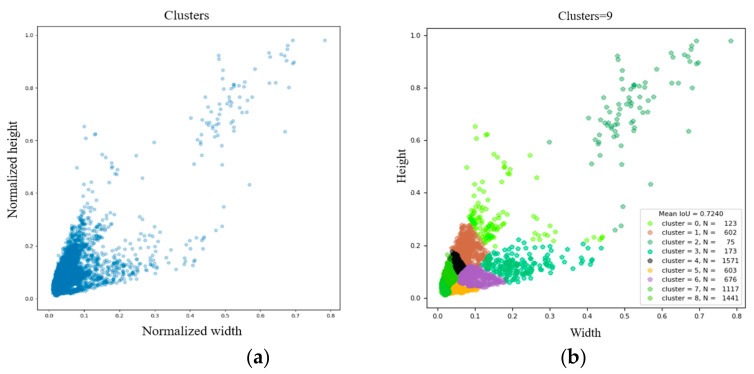
K-means clusters result: (**a**) the size distribution of the drones on the UAV dataset labeled boxes; (**b**) the clustering result of the anchor box.

**Figure 3 micromachines-13-02199-f003:**
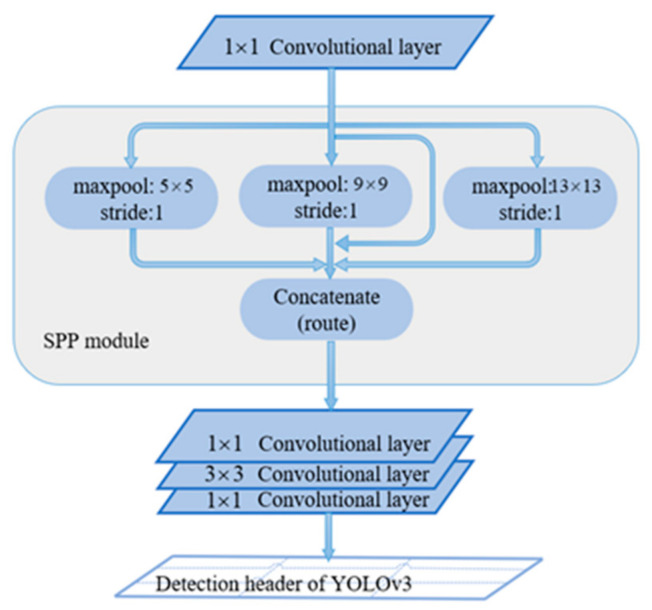
Architecture of the SPP module used in YOLOv3-SPP3.

**Figure 4 micromachines-13-02199-f004:**
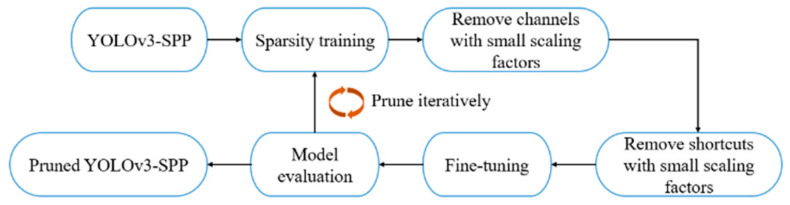
Improved YOLOv3 model cropping flow chart.

**Figure 5 micromachines-13-02199-f005:**
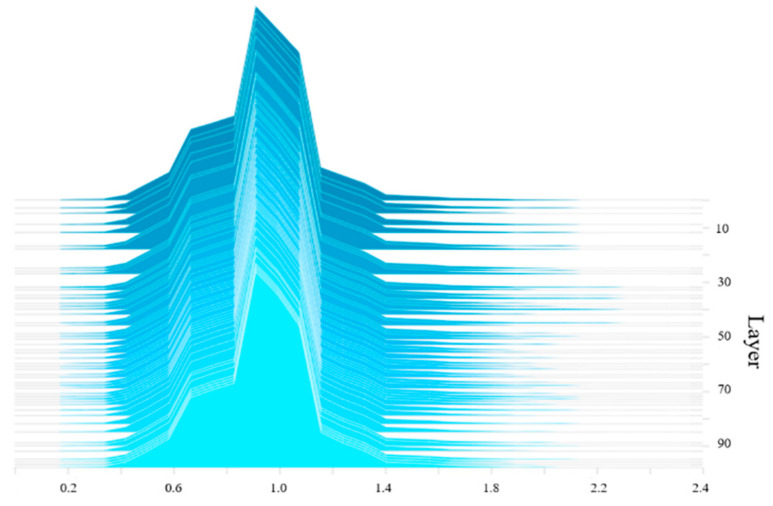
BN layer *γ* value distribution before the sparse training.

**Figure 6 micromachines-13-02199-f006:**
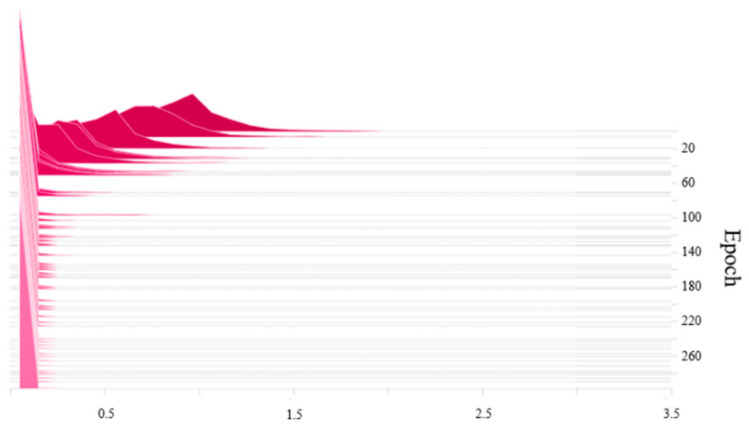
BN layer *γ* value distribution after the sparse training.

**Figure 7 micromachines-13-02199-f007:**
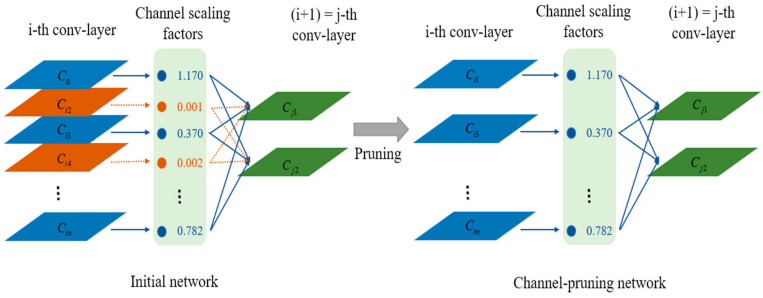
Pruning schematic diagram of the channel pruning algorithm.

**Figure 8 micromachines-13-02199-f008:**
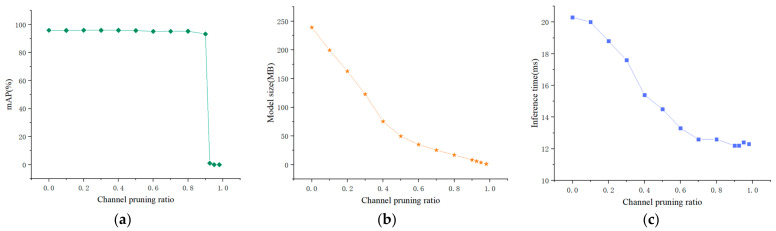
The value of the mAP score, model size, and inference time under different channel pruning rates. (**a**) Model mAP curve. (**b**) Dimension size curve. (**c**) Inference time curve.

**Figure 9 micromachines-13-02199-f009:**
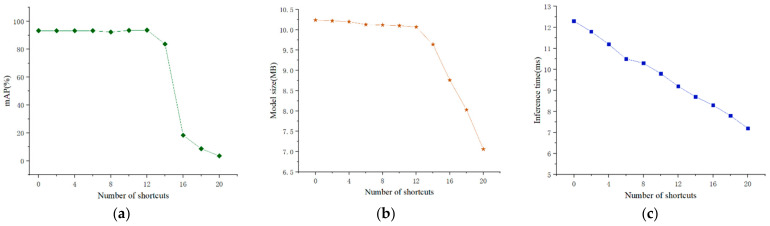
The value of the mAP score, model size, and inference time under a different number of shortcuts. (**a**) Model mAP curve. (**b**) Dimension size curve. (**c**) Inference time curve.

**Figure 10 micromachines-13-02199-f010:**
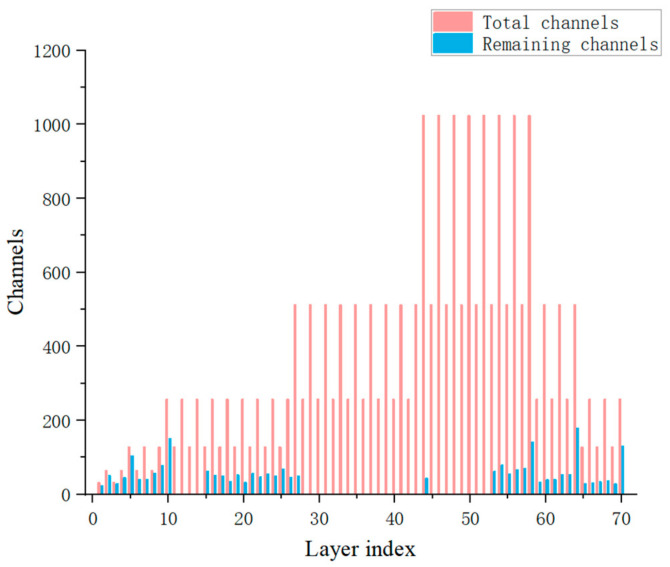
Channel changes before and after pruning.

**Figure 11 micromachines-13-02199-f011:**
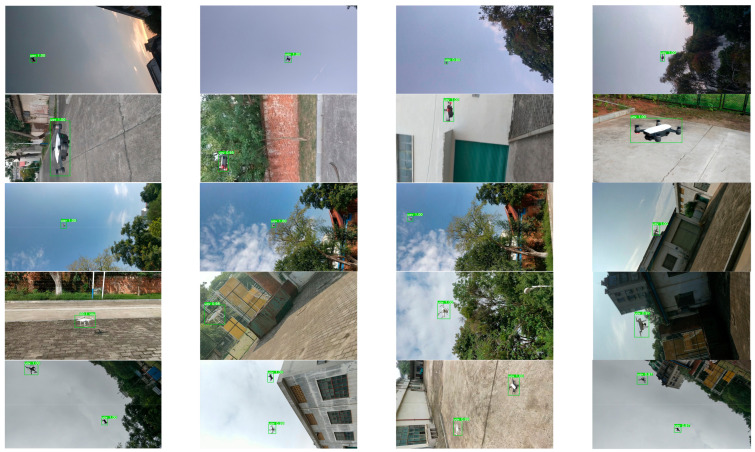
UAV detection results in different environments.

**Table 1 micromachines-13-02199-t001:** Information on the UAV datasets.

Resolution	1920 (Pixels) × 1080 (Pixels)	
Training set (6790 images)	DJI spark	2753 images
	DJI phantom	2652 images
	both DJI spark and phantom	1385 images
Test set (1697 images)	DJI spark	688 images
	DJI phantom	663 images
	both DJI spark and phantom	346 images

**Table 2 micromachines-13-02199-t002:** Parameter settings for the YOLOv3-SPP3 pruning.

Stage	Parameters	Value
Sparse training	Learning rate	0.001
	Batch size	20
	Epochs	300
Channel and layer pruning Fine-tuning model	Layer keep	0.01
	Batch size	32
	Epochs	50

**Table 3 micromachines-13-02199-t003:** Evaluation results of the channel pruning model.

Channel Pruning Ratio	0	0.1	0.2	0.3	0.4	0.5	0.6	0.7	0.8	0.9	0.925	0.95	0.98
mAP (%)	95.92	95.81	95.89	95.87	95.91	95.70	95.14	95.17	95.22	93.24	1.01	0	0
model size (MB)	239.11	199.48	162.77	122.74	75.33	49.62	35.13	25.22	16.88	10.24	7.01	4.05	2.16
inference time (ms)	18.3	18	16.8	15.6	13.4	12.5	11.3	10.6	10.6	10.2	10.2	10.4	10.3

**Table 4 micromachines-13-02199-t004:** Evaluation results of the layer pruning model.

Number of Shortcuts	0	2	4	6	8	10	12	14	16	18	20
mAP (%)	93.24	93.24	93.23	93.25	93.28	93.47	93.66	83.64	18.35	8.65	3.55
model size (MB)	10.24	10.22	10.20	10.13	10.12	10.10	10.07	9.64	8.76	8.03	7.06
inference time (ms)	12.5	11.8	11.2	10.5	10.3	9.8	9.2	8.7	8.3	7.8	7.3

**Table 5 micromachines-13-02199-t005:** Evaluation results before and after pruning.

Parameters	Original Model	After Pruning	After Fine Tuning
mAP (%)	95.92	83.64	95.15
Model size (MB)	243	9.64	11.77
Inference time (ms)	18.3	8.7	8.9

**Table 6 micromachines-13-02199-t006:** UAV detection results obtained by different object detection algorithms.

Algorithms	Precision (%)	Recall (%)	mAP (%)	Model Size (MB)	Detection Speed (f/s)
YOLOv3	84.26	95.51	95.27	235.52	56
Tiny-YOLOv3	74.15	80.26	79.95	33.79	145
CenterNet	93.48	97.75	96.21	237.01	38
SSD300	73.66	86.79	85.42	91.45	49
Faster R-CNN	97.12	98.23	98.96	568	11
Ours	90.20	95.35	95.15	11.77	112

## Data Availability

Data are only available upon request due to restrictions regarding, e.g., privacy and ethics. The data presented in this study are available from the corresponding author upon request. The data are not publicly available due to their relation to other ongoing research.

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
