# Peer review of "Real-Time Detection of Drones Using Channel and Layer Pruning, Based on the YOLOv3-SPP3 Deep Learning Algorithm"

_micromachines, 2022, doi:10.3390/mi13122199_

Round 1
Reviewer 1 Report
This paper presents a real-time detection model of UAV based on YOLOV3-SSP3, and constructs the system for testing. Compared with the traditional detection methods, the detection accuracy and speed have been significantly improved.
This article has clear ideas, compact structure and beautiful and concise diagrams. However, the author must consider the following suggestions to further improve the quality of his work.
1. Since this paper is about achieving detection and tracking for a specific target of UAVs, the importance of UAV detection should be described in more detail in the introduction section to show the importance of this work.
2. In the description of the current state of research, the existing research status should be described in more detail so that the reader can understand the advantages and disadvantages of other algorithms.
3. YOLOv3 is not the latest model of YOLO series. Whether other models of YOLO series should be added in the part of model detection performance comparison makes the experimental results more convincing.
Author Response
Reply
Dear Editor and Reviewers:
Thanks for your letter and the reviewers’ comments on our manuscript entitled “Real-time detection of drones using channel and layer pruning based on YOLOv3-SPP3 deep learning algorithm” (ID: micromachines-2086289). These comments are all valuable and very helpful for revising and improving our paper, as well as have important guiding significance for our research. We have carefully considered all comments from the reviewers and revised our manuscript accordingly. The manuscript has also been double-checked, and the typos and grammar errors we found have been corrected. Below are our individual responses to each of the reviewers' comments. We believe that our responses have well addressed all concerns from the reviewers. We hope our revised manuscript can be accepted for publication.
Q1: Since this paper is about achieving detection and tracking for a specific target of UAVs, the importance of UAV detection should be described in more detail in the introduction section to show the importance of this work.
A1: We have re-added the content about the importance of UAV detection in lines 33-43 of Introduction, and cited two references, reference [1] and reference [2], to illustrate the importance of our work on UAV identification.
[1]. Ravil I. Mukhamediev; Adilkhan Symagulov; Yan Kuchin; Elena Zaitseva; Alma Bekbotayeva; Kirill Yakunin; Ilyas Assanov; Vitaly Levashenko; Yelena Popova; Assel Akzhalova; Sholpan Bastaubayeva; Laila Tabynbaeva. Review of Some Applications of Unmanned Aerial Vehicles Technology in the Resource-Rich Country. Applied Sciences, 2021, 11, pp.10171.
[2]. Noor, N.M.; Abdullah, A.; Hashim, M. Remote sensing UAV/drones and its applications for urban areas: A review. IOP Conf. Ser. Earth Environ. Sci. 2018, 169, pp.012003.
Q2: In the description of the current state of research, the existing research status should be described in more detail so that the reader can understand the advantages and disadvantages of other algorithms.
A2: In order to describe the existing research status in more detail, the disadvantages of traditional target detection algorithms are introduced in lines 66-69, and the advantages and disadvantages of the cited two-stage algorithms SPP-Net, Fast RCNN and Faster RCNN are introduced in lines 81-89. The advantages and disadvantages of YOLOV1, YOLOV2, and SSD algorithms are introduced in lines 98-103. In line 56 we have added a new reference on the acoustic positioning of drones. The modification is as follows:
In lines 66-69:The traditional target detection algorithm based on manual feature extraction mainly has some shortcomings, such as poor recognition effect, low accuracy, large amount of computation, slow operation speed, and may produce multiple correct recognition results.
在第 81-89 行:SPP 需要训练 CNN 提取特征,然后训练 SVM 对这些特征进行分类,这需要巨大的存储空间,而且多阶段训练过程非常复杂。此外,SPP-net仅微调全连接层,而忽略网络其他层的参数。Fast RCNN仍然选择选择性搜索算法来查找感兴趣的区域,这通常是一个缓慢的过程。尽管Faster RCNN具有更高的精度,更快的速度并且非常接近实时性能,但在随后的检测阶段仍然具有一定的计算冗余。此外,如果IOU阈值设置得很低,则会导致噪声检测;如果IOU阈值设置得很高,则会导致过度拟合。
在第98-103行:与两阶段目标检测算法相比,YOLOv1算法的检测速度虽然有了很大的提高,但精度相对较低。YOLOv2算法只有一个检测分支,网络缺乏对多尺度上下文信息的捕获,因此对不同大小目标的检测效果仍然较差,尤其是对于小目标检测。SSD算法检测速度慢,无法满足实时检测的要求。
引用文献:[8]。刘汉森, 范宽刚*, 何兵.使用多层感知器的无人机声学定位。应用人工智能, 2021.05,35(7), 页码: 537-548.
问3:YOLOV3不是YOLO系列的最新型号。在模型检测性能对比部分是否应加入YOLO系列其他型号,使实验结果更具说服力。
答3:非常感谢您的提问。我们的研究目前基于YOLOv3。目前没有基本的更高版本进行研究。由于后续版本的YOLOv3并没有放弃YOLOv3的原有网络,仍然采用骨干网络的整体网络结构结合多尺度特征提取网络,YOLOv3仍然具有较高的研究价值。我们将根据您的意见跟进基于更高版本的研究,以提高识别和检测的性能。模型构建是性能改进的关键,不同的版本仅用于微调性能。我们最近的研究与YOLOv5有关,最新结果将出现在我们的下一篇论文中。
最后,感谢您在百忙之中对我们论文的评论。我们已根据您的意见对文件进行了修改。您的意见大大改进了我们的论文。

Reviewer 2 Report
By removing YOLOv3-16 SPP3's residual structures and convolutional channels, the authors suggested a method for drone detection.
Technically, this article is good. It only needs a revision in the presentation. My recommendation is to accept the paper with revision. Fingers crossed.
1- Please write the name of the tool/environment you used for evaluation and the number of percentages, and the methods you used for comparison in the abstract, introduction, and conclusion sections.
2- The motivation and contribution of the work are not clear at all. The introduction section needs significant revision.
3- Where is the related work section? Please discuss the works in the literature. Section two should be related work section.
4- The organization of this work is inferior. Please read the other articles and reorganize the sections. I suggest the authors reorganize the proposed approach and the performance evaluation sections. My suggestion is as follows:
3. Proposed approach
3.1. Reference architecture (System Model)
3.2. Problem Formulation
3.3. Proposed task offloading approach
4. Performance evaluation
4.1. Simulation setup
4.1.1. Simulation metrics
4.1.2. Simulation scenarios
4.2. Experimental results
You can inspire by my suggestion to reorganize these sections. It is only an example.
5- The conclusion section should be one paragraph. Moreover, it needs to include what you should say in the conclusion section. Please revise it.
6- I highly recommend the authors talk about the Internet of Things (IoT) and the Internet of Drones (IoD) in the paper. The reference architecture is a proper place. Please cite the following articles in the article and talk about drones in the IoD and IoT.
6-1 "FPFTS: a joint fuzzy particle swarm optimization mobility‐aware approach to fog task scheduling algorithm for Internet of Things devices." Software: Practice and Experience 51.12 (2021): 2519-2539.
6-2 "Applications, deployments, and integration of internet of drones (iod): a review." IEEE Sensors Journal (2021).
6-3 "Survey on collaborative smart drones and internet of things for improving smartness of smart cities." Ieee Access 7 (2019): 128125-128152.
Please cite 6-1 for describing IoT applications and architecture. Moreover, please cite 6-2 and 6-3 for describing drones in the IoT and IoD.
Author Response
Reply
Dear Editor and Reviewers:
Thanks for your letter and the reviewers’ comments on our manuscript entitled “Real-time detection of drones using channel and layer pruning based on YOLOv3-SPP3 deep learning algorithm” (ID: micromachines-2086289). These comments are all valuable and very helpful for revising and improving our paper, as well as have important guiding significance for our research. We have carefully considered all comments from the reviewers and revised our manuscript accordingly. The manuscript has also been double-checked, and the typos and grammar errors we found have been corrected. Below are our individual responses to each of the reviewers' comments. We believe that our responses have well addressed all concerns from the reviewers. We hope our revised manuscript can be accepted for publication.
Q1: Please write the name of the tool/environment you used for evaluation and the number of percentages, and the methods you used for comparison in the abstract, introduction, and conclusion sections.
A1: In lines 19-25, 138-141, and 411-428, we have written the experimental equipment and environment used in the article in the abstract, introduction, and conclusion, respectively, and have quantified and collated the experimental results and written the five algorithms used in the comparative experiments conducted in this paper. The modifications made are as follows:
In lines 19-25: The experimental results obtained by using Ubuntu server under python3.6 environment show that the YOLOv3-SPP3 algorithm is better than YOLOv3, Tiny-YOLOv3, Center Net, SSD300 and Faster R-CNN. There is significant compression in size, the maximum compression factor is 20.1 times, the maximum detection speed is increased by 10.2 times, the maximum map value is increased by 15.2%, and the maximum Precision is increased by 16.54%. The proposed algorithm achieves the mAP score of 95.15% and the detection speed of 112f/s, which can meet the requirements of real-time detection of UAVs.
In lines 138-141: Experimental results show that compared with five reference algorithms, namely YOLOv3, Tiny-YOLOv3, Center Net, SSD300 and Faster R-CNN, it has good performance in Precision, Recall, Model size and Detection speed, and has excellent characteristics such as real-time, rapidity and accuracy.
In lines 411-428:To achieve an accurate and real-time detection of drones in natural environments, this research proposed a drone detection method based on the YOLOV3 and pruning algorithms. An accurate and rapid detection of drones was realized using the YOLOv3-SPP3 algorithm. The proposed method reduces the number of learnable parameters by pruning the YOLOv3-SPP3 channel and layers. Under the condition that the detection accuracy of the model is kept almost unchanged, the model size and inference time of the proposed drone detection model was reduced by 95.16% and 51.37%.The experimental results obtained with Ubuntu server in python 3.6 environment show that the proposed method reduces the model size to 1/20 compared with YOLOv3 and doubles the detection speed; improves the map value by 15.2% compared with Tiny-YOLOv3 and reduces the model size to 1/3 compared with Tiny-YOLOv3;improves the detection speed by 3 times compared with Center Net Compared with Center Net, the detection speed is increased by 3 times, and the model size is reduced to 1/20 of the original size; compared with SSD300, the model size is reduced to 1/8 of the original size, and the detection speed is increased by 2.3 times; compared with Faster R-CNN, the model size is reduced to 1/28 of the original size, and the detection speed is increased by 10 times. The proposed method can effectively simplify the detection model and realize an accurate and real-time detection of drones in natural environments.
Q2: The motivation and contribution of the work are not clear at all. The introduction section needs significant revision.
A2: In lines 130-138 we describe the work and contributions we have made and have modified the original introductory section. The changes are as follows:
In lines 130-138:In order to solve the problems of low accuracy and low efficiency in detection and recognition of small targets of unmanned aerial vehicles (UAVS), we proposed a fast identification and detection method for UAVs based on pruning of YOLOv3-SPP3 convolution channel and residual structure (YOLOv3-SPP3 algorithm). Firstly, the K-means algorithm was used to cluster the label boxes, and then the channel pruning and layer pruning algorithms were used to prune the model. Finally, the model was fine-tuned. YOLOv3-SPP3 uses Darknet-53 as its backbone network, which has 23 fast network layers. The model is further compressed by trimming the unimportant convolution channels and the fast network layer.
Q3: Where is the related work section? Please discuss the works in the literature. Section two should be related work section.
A3: In the second part of the article, Related Works, we introduce our related work and modify the title of the second chapter. A brief summary of what we did was added in lines 143-146. The changes are as follows:
In line 142: We have changed the original 2. Materials and Methods to 2. Related Works;
In lines 143-146:In order to achieve the detection and recognition of small UAVs, we first carried out the preparation of the UAV dataset, and after the dataset was annotated, the sample features of the UAV dataset were re-clustered using the K-means clustering algorithm; and the YOLOv3-SPP3 algorithm was proposed.
In line 148: We have changed A. Data Acquisition to 2.1 UAV image acquisition;
In line 170: We have changed B. K-means Clusters to 2.2 K-means clustering of UAV datasets;
In line 183: We have changed C. YOLOv3-SPP3 to 2.3 Model presentation-YOLOv3-SPP3.
Q4: The organization of this work is inferior. Please read the other articles and reorganize the sections. I suggest the authors reorganize the proposed approach and the performance evaluation sections. My suggestion is as follows:
- Proposed approach
3.1. Reference architecture (System Model)
3.2. Problem Formulation
3.3. Proposed task offloading approach
- Performance evaluation
4.1. Simulation setup
4.1.1. Simulation metrics
4.1.2. Simulation scenarios
4.2. Experimental results
You can inspire by my suggestion to reorganize these sections. It is only an example.
A4:As for the layout structure of Chapter 3 and Chapter 4 of this paper, we have modified it according to the content of the article and the structure of other articles. The changes are as follows:
- Method and Performance evaluation
3.1 Sparse training
3.2 Channel and layer pruning
3.2.1 Channel pruning
3.2.2 Layer pruning
3.3 Model Fine-tuning
- Experiment and Results
4.1 Evaluation indicators
4.2 Comparison of different object detection algorithms
Q5: The conclusion section should be one paragraph. Moreover, it needs to include what you should say in the conclusion section. Please revise it.
A5:In lines 411-428, we have revised the conclusion into a single paragraph. In the conclusion, the advantages of the proposed algorithm compared with the other five algorithms are introduced in detail, and the experimental results are summarized. The specific modifications are as follows:
In lines 411-428:To achieve an accurate and real-time detection of drones in natural environments, this research proposed a drone detection method based on the YOLOv3 and pruning algorithms. An accurate and rapid detection of drones was realized using the YOLOv3-SPP3 algorithm. The proposed method reduces the number of learnable parameters by pruning the YOLOv3-SPP3 channel and layers. Under the condition that the detection accuracy of the model is kept almost unchanged, the model size and inference time of the proposed drone detection model was reduced by 95.16% and 51.37%.The experimental results obtained with Ubuntu server in python 3.6 environment show that the proposed method reduces the model size to 1/20 compared with YOLOv3 and doubles the detection speed; improves the map value by 15.2% compared with Tiny-YOLOv3 and reduces the model size to 1/3 compared with Tiny-YOLOv3; improves the detection speed by 3 times compared with Center Net Compared with Center Net, the detection speed is increased by 3 times, and the model size is reduced to 1/20 of the original size; compared with SSD300, the model size is reduced to 1/8 of the original size, and the detection speed is increased by 2.3 times; compared with Faster R-CNN, the model size is reduced to 1/28 of the original size, and the detection speed is increased by 10 times. The proposed method can effectively simplify the detection model and realize an accurate and real-time detection of drones in natural environments.
Q6: I highly recommend the authors talk about the Internet of Things (IoT) and the Internet of Drones (IoD) in the paper. The reference architecture is a proper place. Please cite the following articles in the article and talk about drones in the IoD and IoT.
A6:In reference 39, references mentioned in 6-1 is cited. References 6-2 and 6-3 are cited in Reference 40 and 41.
[39]. Saeed Javanmardi, Mohammad Shojafar, Valerio Persico, Antonio Pescapè. FPFTS: A joint fuzzy particle swarm optimization mobility-aware approach to fog task scheduling algorithm for Internet of Things devices. Software: Practice and Experience, 2021, Vol.51(12): 2519-2539.
[40]. Laith Abualigah; Ali Diabat;Putra Sumari;Amir H. Gandomi.Applications, Deployments, and Integration of Internet of Drones (IoD): A Review. IEEE Sensors Journal, 2021, Vol.21(22): 25532-25546.
[41]. Saeed H. Alsamhi; Ou Ma; Mohammad Samar Ansari; Faris A. Almalki.Survey on Collaborative Smart Drones and Internet of Things for Improving Smartness of Smart Cities. IEEE Access, 2019, Vol.7: 128125-128152.
Finally, thank you for your comments on our paper during your busy schedule. We have made changes to the paper based on your comments. Your comments have greatly improved our paper.

Round 2
Reviewer 1 Report
26/2000
The graph in the paper needs to be introduced with the coordinate axis,
Reviewer 2 Report
The article is now ok. Please check that the references are correct before submitting the article's last version to the journal. If you used Latex/Sharelatex, please use BibTeX.
Fingers crossed.